# ALL YOU NEED ARE RANDOM VISUAL TOKENS? DEMYSTIFYING TOKEN PRUNING IN VLLMS

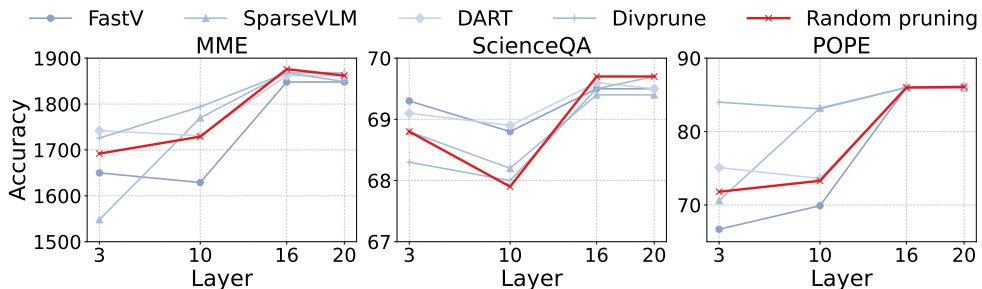

Figure 1: **Existing token pruning methods exhibit similar performance to random selection at deeper layers.** We compare various pruning methods on LLaVA-1.5-7B model and three benchmarks, with 90% of visual tokens are pruned within a given language decoder layer.

## ABSTRACT

Vision Large Language Models (VLLMs) usually incur high computational costs due to their reliance on hundreds of visual tokens to represent images. While token pruning offers a promising solution for accelerating inference, this paper, however, identifies a key observation: in deeper layers (*e.g.*, beyond the 20th), existing training-free pruning methods *perform no better than random pruning*. We hypothesize that this degradation is caused by **"vanishing token information"**, where visual tokens progressively lose their salience with increasing network depth. To validate this hypothesis, we formally quantify a token's information content by measuring the perturbation to the model's output probability upon its removal. Using this proposed metric, our analysis of the information of visual tokens across layers reveals three key findings: (1) As layers deepen, the information of visual tokens gradually becomes uniform and eventually vanishes at an intermediate layer, which we term as "information horizon", beyond which the visual tokens become redundant; (2) The position of this horizon is not static; it extends deeper for visually intensive tasks, such as Optical Character Recognition (OCR), compared to more general tasks like Visual Question Answering (VQA); (3) This horizon is also strongly correlated with model capacity, as stronger VLLMs (*e.g.*, Qwen2.5-VL) make more effective use of deeper visual tokens compared with weaker models (*e.g.*, LLaVA-1.5). Based on our findings, we show that simple random pruning in deep layers efficiently balances performance and efficiency. Moreover, integrating random pruning consistently enhances existing methods across various models and benchmarks, with improvements up to 6.7% on LLaVA-1.5-7B. Using DART with random pruning achieves state-of-the-art results, maintaining 93.9% of Qwen-2.5-VL-7B performance while pruning 50% of visual tokens.

## 1 INTRODUCTION

Vision Large Language models (VLLMs) (Bai et al., 2025; Li et al., 2024; Chen et al., 2025; Liu et al., 2023; Zhu et al., 2023) have achieved remarkable success on a wide range of multi-modal

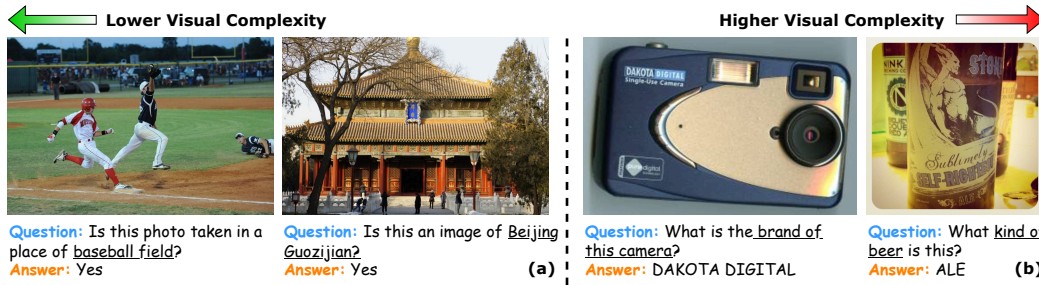

Figure 2: **Tasks with different visual complexity.** Low visual complexity tasks only require the VLLM to identify global information such as the main scene, while tasks with higher visual complexity require the model to focus on visual details.

tasks including knowledge QA (Lu et al., 2022; Fu et al., 2024; Chen et al., 2024), spatial reasoning (Rajabi & Kosecka, 2024; Zhang et al., 2024; Cheng et al., 2024) and OCR (Singh et al., 2019; Mishra et al., 2019; Liu et al., 2024d), by integrating a visual encoder with the large language model. However, VLLMs usually convert the image into extensive visual tokens, which dominate the input sequence length and significantly slow down the inference process.

Reducing the number of visual tokens is therefore critical for efficient VLLM deployment, and various training-free token pruning strategies have been explored, which can be classified into two main categories according to their criteria for selecting visual tokens for pruning: (1) importance-based methods, which determine the token importance via attention weights and discard tokens deemed less important (Chen et al., 2024; Zhang et al., 2025b; Ye et al., 2025; Liu et al., 2024b; Zhao et al., 2025); (2) diversity-based methods, which assess the similarity between tokens and remove the redundant ones (Bolya et al., 2023; Alvar et al., 2025; Wen et al., 2025; Wang et al., 2025). These techniques exhibit promising performance in significantly reducing inference cost while maintaining the majority of VLLMs' performance. However, we observe a notable limitation: in deep layers of VLLMs' language decoder, existing token pruning methods perform **similarly or even worse than random pruning** (*i.e.*, randomly selecting tokens to remove). As shown in Figure 1, we employ various token pruning methods at different layers of the LLaVA-1.5-7B model (Liu et al., 2024a). In the deeper layers (*e.g.*, 16th to 20th), none of the evaluated pruning methods show better performance than random pruning across three benchmarks (Fu et al., 2024; Lu et al., 2022; Li et al., 2023). These findings lead to an intriguing question: *when performing no better than random, can these pruning methods identify visual tokens containing information necessary to produce the answer?*

To answer this question, we firstly propose to estimate a visual token's information by measuring changes in model output probabilities when this token is removed. Specifically, as illustrated in Figure 3, we initially prune all visual tokens except the target one at a specific layer and calculate the model's predicted probability on the ground-truth label. Subsequently, we further remove this visual token at the same layer, forcing the model to rely solely on text tokens. Finally, the difference in probabilities with and without the target visual token serves as the estimate of its information at the specified layer. Experimental evidence illustrates the effectiveness of the proposed information modeling: removing low-information tokens based on our definition consistently improves the model's performance.

From the lens of visual token information, we rethink the failure of existing token pruning methods in deep layers. Our experimental results demonstrate that: most pruning techniques effectively retain visual tokens with high information up to the 10th layer of the LLaVA-1.5-7B model. Beyond the 14th layer, however, random pruning similarly maintains visual token information. The primary cause is that visual token information vanishes uniformly as layers deepen, ultimately reaching zero beyond a specific layer (see Figure 6), which we term as "*information horizon*". In fact, the visual tokens after the information horizon could be entirely removed without compromising the model's performance. We further illustrate that the position of the information horizon is dynamic, influenced by two primary factors. (1) Task visual complexity: comparing to tasks where the model simply answers knowledge questions based on objects (Fu et al., 2024; Lu et al., 2022) (see Figure 2(a)), the tasks demanding more visual cues, as shown Figure 2(b) (Singh et al., 2019; Mishra et al.,

2019; Liu et al., 2024d), depend on deeper layer visual tokens; (2) Model visual capability: for the same task, the models with superior visual abilities (*e.g.*, Qwen-2.5-VL-7B (Bai et al., 2025)) can exploit deeper layers visual tokens than less advanced models like LLaVA-1.5-7B (Liu et al., 2024a), extending its information horizon to deeper layers.

Due to the dynamic information horizon, removing all visual tokens at a fixed layer could hinder performance in tasks with higher visual complexity or for models with enhanced visual abilities. In this paper, we demonstrate that simply integrating random pruning with existing pruning methods can more effectively balance inference efficiency and accuracy across various datasets. For example, in the case of Qwen-2.5-VL-7B, combining random pruning with DART enhances performance on OCRBench from 75.5% to 77.9%, maintaining 93.9% of the initial model performance while removing 50% visual tokens. For LLaVA-1.5-7B, employing DivPrune + Random pruning results in a 6.7% improvement on MMBench compared to using only DivPrune (61.3% *vs.* 54.6%). Our contributions are summarized as follows:

- We propose to quantify the visual token information in VLLMs by calculating the change in output probabilities, demonstrating that removing low-information visual tokens improves the model's performance.

- We observe that the information in visual tokens gradually becomes uniform and eventually disappears at an intermediate layer (information horizon). Post this layer, visual tokens can be discarded without affecting model performance.

- We demonstrate that both task visual complexity and model visual capability impact the position of information horizon, illustrating that integrating random pruning with existing pruning methods could be more effectively balance the performance and efficiency.

## 2 RELATED WORK

**Visual Large Language Model.** Visual Large Language Models (VLLMs) demonstrate impressive capabilities, showing strong performance on diverse multimodal tasks (Huang et al., 2025; Li et al., 2024) including (i) perception & grounding (Fu et al., 2024; Ma et al., 2024; Li et al., 2023), (ii) knowledge-intensive multimodal reasoning (Lu et al., 2022; Hudson & Manning, 2019; Fu et al., 2025), and (iii) text-centric understanding that requires reading in-image text (Singh et al., 2019; Liu et al., 2024d; Mathew et al., 2021). A typical VLLM comprises three components: (i) a vision encoder, (ii) a vision–language connector, and (iii) a pretrained large language model for reasoning and generation. While this architecture has become the dominant paradigm, it incurs high computational costs as VLLMs often encode images into numerous tokens. For example, LLaVA-1.5 encodes a $336 \times 336$ image into 576 tokens (Liu et al., 2024a), while Qwen2.5-VL can produce thousands of tokens for high-resolution images (Bai et al., 2025). Long visual sequences introduce substantial computational overhead, urging further research on sparsification to reduce redundancy while preserving essential information for downstream tasks.

**Visual Token Pruning.** To mitigate the computational overhead introduced by long visual sequences, recent works explore *visual token pruning* as an efficient acceleration strategy. Existing training-free visual token pruning methods can be broadly divided into two categories: *importance-based* and *diversity-based*. **Importance-based methods** rely on task-driven importance signals, typically guided by attention (Chen et al., 2024; Zhang et al., 2025b; Zhao et al., 2025; Ju et al., 2024; Xing et al., 2024). FastV (Chen et al., 2024) exploits the attention of the final text token, which directly determines the next output token, as a principled signal to eliminate redundant visual tokens, while SparseVLM (Zhang et al., 2025b) adopts a two-stage strategy by first selecting salient text tokens and subsequently leveraging their attention to more precisely guide visual token pruning. SGL (Zhao et al., 2025) introduces a small-to-large guidance paradigm, where global attention aggregated from a small VLLM is used to guide token pruning in a large VLLM. PDrop (Xing et al., 2024) performs pruning progressively across layers leverages the attention of the final text token. Despite their effectiveness, importance-based methods inherently rely on attention maps, making them incompatible with FlashAttention (Dao et al., 2022; Dao, 2024). **Diversity-based methods** typically calculates similarity between visual tokens and removes redundant tokens, offering efficient acceleration compatibility (Bolya et al., 2023; Alvar et al., 2025; Wen et al., 2025; Zhang et al., 2025a; Li et al., 2025). For example, DART (Wen et al., 2025) prunes redundant tokens by explicitly

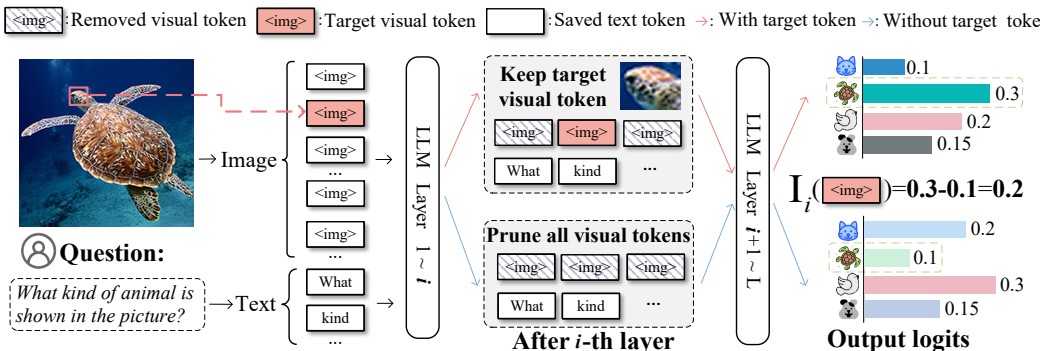

Figure 3: **Illustration of our framework for computing visual token information**. At the $i$-th layer of VLLM's language decoder, we firstly remove all other visual tokens except the target one and run one forward pass. Next we additionally run another forward pass by further removing the only one visual token. The difference between these two output probabilities on the ground-truth label defines the information score $I_i(\mathbf{V}_k)$.

measuring duplication, and DivPrune (Alvar et al., 2025) formulates token pruning as a Max–Min Diversity Problem to maximize the diversity of retained tokens. Although both categories strike a reasonable balance between inference speed and accuracy, we observe that, as depicted in Figure 1, pruning at deeper layers can lead to performance similar to random pruning. This motivates our investigation into the role and characteristics of visual tokens at deep layers.

## 3 METHODOLOGY

In this section, we first provide an overview of VLLM in Section 3.1, and introduce the definition of **visual token information** in Section 3.2. Subsequently, Section 3.3 demonstrates the effectiveness of the proposed information modeling via experimental evidence.

### 3.1 PRELIMINARY: VISION LARGE LANGUAGE MODEL

Current mainstream architectures of Vision Large Language Models (VLLMs) follow a common design paradigm, consisting of a visual encoder, a modality projector, and a language decoder. Given an input image $\mathbf{X}_v$, a pretrained visual encoder (*e.g.*, CLIP (Radford et al., 2021) or SigLIP (Zhai et al., 2023) vision encoder) is used to extract image features $\mathbf{Z}_v = g(\mathbf{X}_v)$, where $g(\cdot)$ denotes the visual encoder. To align visual and language modalities, a projection module maps $\mathbf{Z}_v$ into visual embeddings $\mathbf{V} \in \mathbb{R}^{d \times N_v}$, where $d$ denotes the embedding dim and $N_v$ is the number of visual tokens, which is fixed as 576 in LLaVA-1.5 (Liu et al., 2024a) or scales based on image resolution in Qwen2.5-VL (Bai et al., 2025). Subsequently, $\mathbf{V}$ is fed into a pre-trained language decoder $f_\theta(\cdot)$, which integrates both visual and textual embeddings to predict the probability of the next generated token in an auto-regressive manner:

$$p(g_i) = f_\theta(\mathbf{V}, \mathbf{T}, \mathbf{G}^{1:i-1}), \quad g_i = \arg\max_x p(x) \tag{1}$$

where $\mathbf{T} \in \mathbb{R}^{d \times N_t}$ represents the embeddings of input textual tokens (*e.g.*, instruction and question), $g_i$ is the $i$-th generated token, and $p(g_i)$ denotes its probability distribution over the vocabulary. $\mathbf{G}^{1:i-1} \in \mathbb{R}^{(i-1) \times d}$ represents the embeddings of previously generated tokens. At each decoding step, the token with the highest predicted probability is appended to the generated sequence, which is then used as input for the next step. Throughout this section, we concentrate on the *prefill* step, which predicts the *first output token* probability as $p(g_1) = f_\theta(\mathbf{V}, \mathbf{T})$.

### 3.2 VISUAL TOKEN INFORMATION

Let $\mathcal{V} = \{1, \ldots, N_v\}$ denote the set of indices for the visual tokens, $y \in \{1, \ldots, |V|\}$ represent the vocabulary index of the first generated token in the ground-truth answer, where $|V|$ is the length of the vocabulary. For a specific visual token index $k \in \mathcal{V}$, we denote the information contributed by

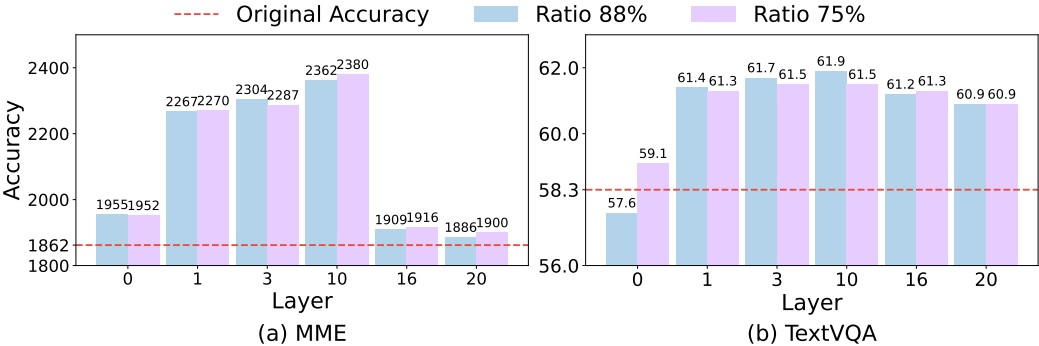

Figure 4: **Effectiveness of proposed visual token information modeling.** Performance of LLaVA-1.5-7B on MME (a) and TextVQA (b) when pruning 88% and 75% visual tokens with low information at different layers. Layer 0 denotes pruning applied between the visual encoder and the language decoder. Notably, removing the low-information visual tokens consistently improves the model's performance across different benchmarks.

$\mathbf{V}_k$ at the $i$-th layer of VLLM's decoder as $\mathrm{I}_i(\mathbf{V}_k)$. As illustrated in Figure 3, to estimate $\mathrm{I}_i(\mathbf{V}_k)$, we measure the change in the probability of the first generated token $g_1$ on $y$ when $\mathbf{V}_k$ is removed at the $i$-th layer.

Specifically, let $\mathbf{H}_v^i, \mathbf{H}_t^i = f_\theta^{1:i}(\mathbf{V}, \mathbf{T})$ denote the hidden states of the visual and text embeddings after the first $i$ layers of the decoder $f_\theta(\cdot)$, and $\mathbf{H}_v^i = \{\mathbf{H}_1^i, \mathbf{H}_2^i, \ldots, \mathbf{H}_{N_v}^i\}$. To isolate the interference from other visual tokens, we first employ a binary mask $\mathbf{M} \in \{0,1\}^{1 \times N_v}$ to remove all visual tokens except $\mathbf{H}_k^i$:

$$M_j = \begin{cases} 1 & \text{if } j = k, \\ 0 & \text{otherwise.} \end{cases} \quad (2)$$

The masked visual representation is $\hat{\mathbf{H}}_v^i = \mathbf{H}_v^i \odot \mathbf{M}$, where $\odot$ denotes element-wise multiplication. Next, we pass the masked hidden states through the remaining decoder layers $f_\theta^{i+1:L}$, where $L$ denotes the total number of decoder layers in the model, to yield a distribution over the vocabulary:

$$p(g_1) = f_\theta^{i+1:L}(\hat{\mathbf{H}}_v^i, \mathbf{H}_t^i). \quad (3)$$

The probability assigned to the ground-truth label $y$ could be represented as:

$$p_k = p(g_1)[y]. \quad (4)$$

Subsequently, to measure the incremental contribution of $\mathbf{V}_k$, we compare $p_k$ with the probability predicted using no visual information after the $i$-th layer. Specifically, we set $\hat{\mathbf{H}}_v^i = \mathbf{0}$ to force the model to rely solely on text hidden states:

$$p_{\text{text}} = f_\theta^{i+1:L}(\mathbf{0}, \mathbf{H}_t^i)[y]. \quad (5)$$

Finally, the information attributed to token $\mathbf{X}_k$ at layer $i$ is defined as:

$$\mathrm{I}_i(\mathbf{V}_k) = p_k - p_{\text{text}}, \quad (6)$$

### 3.3 EFFECTIVENESS OF VISUAL TOKEN INFORMATION

Using the proposed visual token information, we perform extensive pruning experiments using LLaVA-1.5-7B (Liu et al., 2024a), a widely used baseline VLLM that encodes images into a fixed sequence of 576 visual tokens. Specifically, at different layers of VLLM's language decoder, we rank the visual tokens based on their information defined in Equation 6. Subsequently, we remove 75% and 88% of low-information tokens, retaining 144 and 72 visual tokens, respectively. Our experiments are conducted on two popular VQA benchmarks: MME (Fu et al., 2024) and TextVQA (Singh et al., 2019).

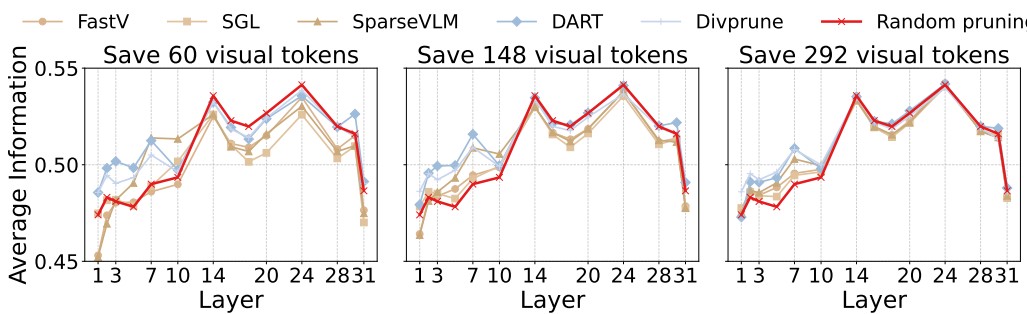

Figure 5: **Evaluation of various pruning methods.** We measure the sum of information in retained visual tokens when using different pruning methods. In the deep layers, existing pruning methods fail to retain more high-information than random pruning.

As illustrated in Figure 4, across two evaluated benchmarks, removing 75% of low-information visual tokens consistently *outperform the base model without pruning*, suggesting that our method effectively quantifies the information contributing to the ground-truth answer in the visual tokens. Notably, our experimental results indicate that low-information visual tokens are unnecessary and may hinder inference. For example, pruning 75% of low-information tokens at the 10th layer leads to 27.8% and 6.1% of performance improvement on MME and TextVQA, respectively. This is possibly due to low-information tokens interfering with the focus of VLLM on tokens with important information.

## 4 EXPERIMENTAL ANALYSIS

### 4.1 UNDERSTANDING TOKEN PRUNING USING INFORMATION

Firstly, we aim to investigate whether current pruning strategies can preserve high-information visual tokens and understand why they may perform no better than random pruning in the deep layers. Specifically, we employ token pruning strategies at different layers of VLLM to remove some visual tokens and measure the sum of information of the remaining ones. Our experiments are conducted on LLaVA-1.5-7B (Liu et al., 2024a) and use 200 randomly selected samples from the MME benchmark (Fu et al., 2024). We evaluate a diverse set of training-free pruning methods including both importance-based and diversity-based methods, including DivPrune (Alvar et al., 2025), FastV (Chen et al., 2024), SparseVLMs (Zhang et al., 2025b), DART (Wen et al., 2025), SGL (Zhao et al., 2025). For each pruning method, we adopt three pruning ratios—90%, 75%, and 50%—corresponding to retaining 60, 148, and 292 visual tokens, respectively.

The evaluation results are summarized in Figure 5. At shallow layers (*e.g.*, 1st to 7th), most pruning methods can more effectively retain high-information visual tokens than random pruning. Notably, across three pruning ratios, diversity-based methods (DivPrune and DART) consistently outperform importance-based methods (SparseVLMs, SGL, FastV), suggesting that diverse visual tokens might carry more information than high-attention-scored ones. After the 10th layer, however, the gap between baseline methods and random pruning gradually narrows. By the 14th layer, random pruning surpasses all baseline methods, and this continues in the subsequent layers, consistent with accuracy trends in deeper layers (as shown in Figure 1). The underlying cause could be that *the visual tokens information uniformly vanish in the deep layers.* As shown in Figure 6, the visual tokens capture varying amounts of information across layers 1 to 7. High-information tokens cluster around key areas, such as the baseball player, whereas low-information tokens provide limited clues about the baseball, like the grass. However, from the 9th layer, the variability in token information starts diminishing. By the 16th layer, the visual tokens uniformly capture negligible information. In this case, the selection of pruned visual tokens does not influence model performance, resulting in similar results for existing pruning methods and random pruning.

Furthermore, when the information of all visual tokens is uniformly low at a specific layer, we can remove **all visual tokens** from this layer without affecting model performance. We refer to this layer

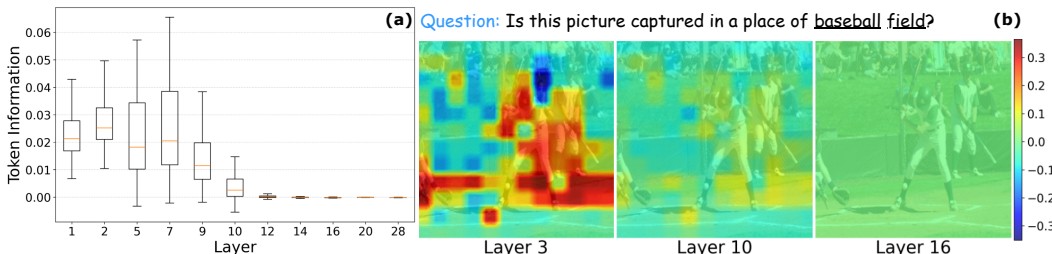

Figure 6: **Visual token information becomes uniform in the deep layers.** (a) The variance of visual token information across different layers of the language decoder. (b) Layer-wise visual token information maps at layers 3, 10, and 16.

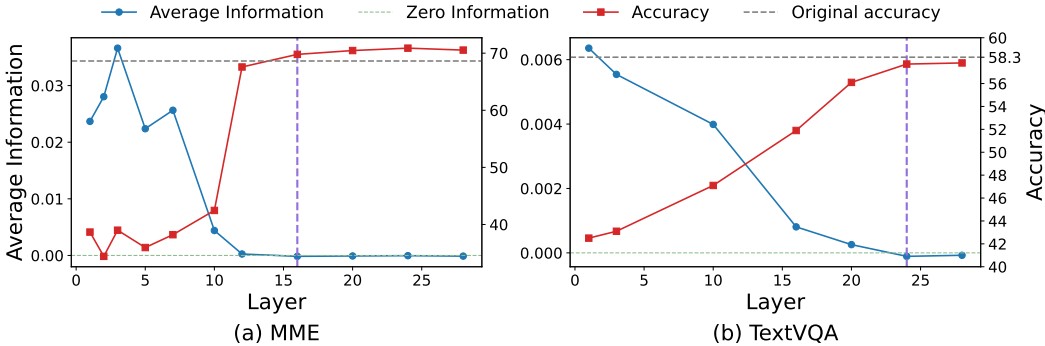

Figure 7: **Information horizon of visual tokens.** The purple vertical dashed line marks the "information horizon". When the mean information of all visual tokens becomes close to zero at a certain layer of LLaVA-1.5-7B, removing these tokens at this layer does not impair the model's performance.

as the "information horizon". As demonstrated in Figure 7, we measure the mean information of all visual tokens across different layers and evaluate the LLaVA-1.5-7B performance when pruning all visual tokens at each layer. For the MME benchmark (Figure 7(a)), at layer 16, the mean information of all visual tokens nearly becomes zero, and removing these tokens at this layer results in performance close to the original model. This phenomenon can also be observed at the 24th layer and TextVQA benchmark (see Figure 7(b)).

### 4.2 POSITION OF THE INFORMATION HORIZON

To further investigate at which layer all visual tokens can be removed without loss of model performance, we conduct comprehensive experiments on Qwen-2.5-VL-7B and LLaVA-1.5-7B across 6 benchmarks, including MME (Fu et al., 2024), ScienceQA (Lu et al., 2022), POPE (Li et al., 2023), TextVQA (Singh et al., 2019), OCRBench (Liu et al., 2024d), and OCRVQA (Mishra et al., 2019). As illustrated in Figure 8, we find there are two primary factors:

**Task visual complexity.** Comparing to knowledge question-answering (Lu et al., 2022) or hallucination detection (Li et al., 2023), more visually complex tasks such as OCR (Liu et al., 2024d) (which depend on precise visual details) rely on deeper visual tokens. For example, with Qwen-2.5-VL-7B (Figure 8(a)), pruning all visual tokens at the 20th layer leads to the highest accuracy for ScienceQA, MME, and POPE, whereas for OCRBench and TextVQA, this occurs around the 27th layer. A similar trend can also be observed on LLaVA-1.5-7B (15th *vs.* 24th layer in Figure 8(b)).

**Model visual capability.** We also compare different VLLMs under the same task. Models with stronger visual understanding, such as Qwen-2.5-VL-7B, are able to exploit informative visual tokens from deeper layers than weaker models like LLaVA-1.5-7B. For example, Qwen-2.5-VL-7B leverages the visual token at the 20th layer, enhancing question-answering precision compared to LLaVA-1.5-7B (96.4% *vs.* 68.6% on MME), with all visual tokens removable without loss from the

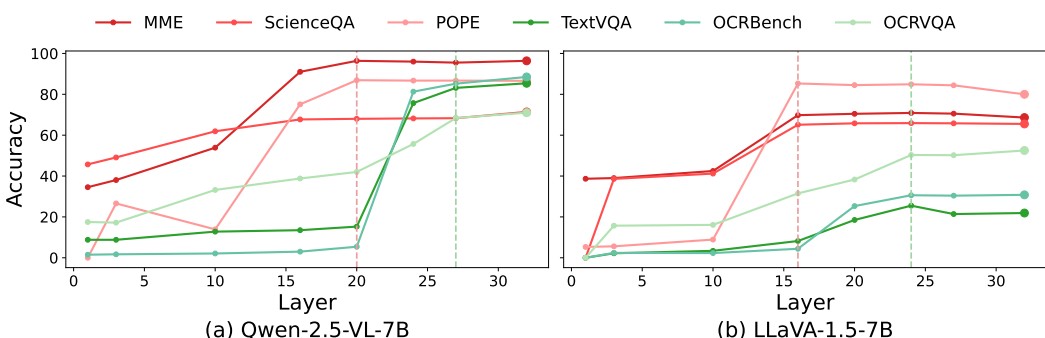

Figure 8: **Performance of models when pruning all visual tokens at different decoder layers**. Colored vertical dashed lines mark the convergence layers for datasets of different visual complexity, beyond which pruning all visual tokens has little impact on performance. This demonstrates that the layer at which visual tokens could be entirely removed depends on task visual complexity and model visual capability.

Table 1: **Performance of Qwen2.5-VL-7B under varying pruning strategies**. Integrating random pruning with existing pruning methods leads to improved performance across multiple benchmarks.

| Method | MME | SQA | VQA$^{\text{Text}}$ | MMB | InfoVQA | DocVQA | OCRBench | Acc. (%) |
|---|---|---|---|---|---|---|---|---|
| *Upper Bound* **(100%)** | | | | | | | | |
| Qwen2.5-VL-7B | 2313 | 71.4 | 85.4 | 79.8 | 82.3 | 94.9 | 88.5 | 100.0 |
| *Retain 50% Tokens* (↓ **50%**) | | | | | | | | |
| DART (Arxiv2025) | 2295 | 69.2 | 82.1 | **79.6** | 64.4 | 88.5 | 75.5 | 92.7 (↓ 7.3) |
| DivPrune (CVPR25) | 2233 | 59.8 | 80.5 | 76 | 67.7 | 88.6 | 73.7 | 89.8 (↓ 10.2) |
| DART+VTW | 2302 | 69.8 | 76.3 | 79.4 | 65.8 | 86.5 | 73.3 | 91.4 (↓ 8.6) |
| DivPrune+VTW | 2243 | 57.9 | 73.4 | 75.6 | 67.9 | 86.3 | 70.9 | 87.5 (↓ 12.5) |
| DART+Random | **2318** | **70.0** | **82.7** | **79.6** | 65.9 | **89.3** | **77.9** | **93.9** (↓ 6.1) |
| DivPrune+Random | 2235 | 59.7 | 80.1 | 75.9 | **68.5** | 88.9 | 73.9 | 89.9 (↓ 10.1) |

16th layer. Moreover, on TextVQA and OCRBench, Qwen-2.5-VL-7B attains approximately 80% accuracy using visual tokens by the 27th layer, whereas LLaVA-1.5-7B remains below 40% beyond the 24th layer.

## 4.3 EFFECTIVENESS OF RANDOM PRUNING

**Combining with random pruning improves existing pruning methods.** As illustrated in Figure 7, tokens in the first 10 layers contain more information than those in deeper layers. In this section, we demonstrate that applying existing pruning techniques to retain high-information tokens in shallow layers, while employing random pruning to eliminate less informative tokens in deeper layers, can better maintain model performance. Specifically, we select three representative pruning techniques, including FastV (Chen et al., 2024), DivPrune (Alvar et al., 2025), and DART (Wen et al., 2025). We also incorporate two multi-layer pruning methods, namely SparseVLM (Zhang et al., 2025b) and PDrop (Xing et al., 2024) for comparison. Experiments are conducted on two VLLM architectures (Qwen-2.5-VL and LLaVA-1.5) across multiple benchmarks with varying visual complexity. More experiment details are provided in Appendix A.

As depicted in Table 1, integrating random pruning enhances DART's performance on OCRBench from 75.5% to 77.9% given the same pruning ratio, maintaining 1.2% more average original Qwen-2.5-VL-7B performance across 7 evaluated benchmarks (92.7% *vs.* 93.9%). Similarly, on LLaVA-1.5-7B, DivPrune + Random pruning achieves 61.3% accuracy on MMBench when pruning 88.9% visual tokens, which is a 6.7% improvement over using only DivPrune (see Table 2). Compared to multi-layer pruning methods like PDrop and SparseVLM, our methods are still competitive: at an 88.9% pruning ratio, combining random pruning with FastV improves LLaVA-1.5-7B performance

Table 2: **Performance of LLaVA-1.5-7B under varying pruning strategies.**

| Method | MME | SQA | POPE | VQA$^{Text}$ | MMB | GQA | MMB-CN | Acc. (%) |
|---|---|---|---|---|---|---|---|---|
| *Upper Bound, 576 Tokens* **(100%)** | | | | | | | | |
| LLaVA-1.5-7B | 1862 | 69.5 | 85.9 | 58.3 | 64.6 | 62 | 58.1 | 100.0 |
| *Retain 192 Tokens* (↓ **66.7%**) | | | | | | | | |
| PDrop (CVPR25) | 1765 | 69.2 | 79.7 | 56.2 | 63.3 | 57.2 | 56.5 | 95.9 (↓ 4.1) |
| SparseVLM (ICML25) | 1779 | 68.6 | 84.9 | **57.7** | 63.4 | 58.9 | 57.1 | 97.6 (↓ 2.4) |
| FastV (ECCV24) | 1796 | 69.1 | 78.3 | 56.4 | 63.2 | 57.7 | 57.4 | 96.2 (↓ 3.8) |
| DART (Arxiv2025) | 1833 | 69.0 | 81.7 | 57.0 | 63.7 | 59.5 | 57.1 | 97.6 (↓ 2.4) |
| DivPrune (CVPR25) | 1769 | 68.6 | **87.6** | 56.6 | 62.3 | **60.1** | 56.2 | 97.6 (↓ 2.4) |
| FastV+VTW | 1848 | 68.6 | 81.8 | 53.6 | **64.7** | 57.5 | 57.6 | 96.7 (↓ 3.3) |
| DART+VTW | 1807 | 69.1 | 84.4 | 56.4 | 64.1 | 59.1 | **57.7** | 97.9 (↓ 2.1) |
| DivPrune+VTW | 1772 | 69.1 | 87.0 | 56.0 | 63.2 | 60.0 | 57.2 | 97.9 (↓ 2.1) |
| FastV+Random | **1853** | 68.6 | 81.5 | 53.5 | **64.7** | 57.8 | 57.5 | 96.7 (↓ 3.3) |
| DART+Random | 1816 | 69.2 | 84.4 | 56.8 | 63.9 | 59.6 | 57.7 | **98.2** (↓ 1.8) |
| DivPrune+Random | 1777 | **69.3** | 87.1 | 55.9 | 63.2 | **60.1** | 56.9 | 97.9 (↓ 2.1) |
| *Retain 64 Tokens* (↓ **88.9%**) | | | | | | | | |
| PDrop (CVPR25) | 1263 | 68.0 | 45.7 | 48.3 | 45.9 | 45.5 | 34.9 | 72.3 (↓ 27.7) |
| SparseVLM (ICML25) | 1487 | 68.7 | 69.9 | 52.6 | 58.6 | 51.9 | 50.3 | 87.3 (↓ 12.7) |
| FastV (ECCV24) | 1088 | 68.2 | 23.0 | 46.0 | 34.1 | 41.4 | 25.9 | 60.9 (↓ 39.1) |
| DART (Arxiv2025) | 1529 | 68.6 | 59.6 | 50.4 | 55.8 | 51.2 | 46.7 | 83.7 (↓ 16.3) |
| DivPrune (CVPR25) | 1614 | 67.9 | 85.6 | **55.0** | 54.6 | 57.7 | 52.8 | 92.4 (↓ 7.6) |
| FastV+VTW | 1361 | 70.4 | 42.1 | 47.0 | 54.6 | 47.1 | 46.4 | 77.8 (↓ 22.2) |
| DART+VTW | 1664 | 68.3 | 74.0 | 53.3 | 60.3 | 55.5 | 53.1 | 91.3 (↓ 8.7) |
| DivPrune+VTW | 1677 | 68.2 | **86.8** | 54.1 | **61.6** | 57.1 | 53.7 | 94.6 (↓ 5.4) |
| FastV+Random | 1350 | **70.6** | 42.1 | 47.4 | 54.9 | 46.6 | 46.4 | 77.8 (↓ 22.3) |
| DART+Random | 1670 | 68.4 | 74.1 | 53.4 | 60.4 | 55.1 | 53.4 | 91.5 (↓ 8.5) |
| DivPrune+Random | **1693** | 68.3 | **86.8** | 54.5 | 61.3 | **57.8** | 53.8 | **94.9** (↓ 5.1) |

by 5.5% over PDrop, while DART + Random pruning enhances average accuracy by 4.2% compared to SparseVLM.

**Random pruning *vs.* visual token withdraw.** Furthermore, we compare the random pruning with withdrawing all visual tokens (denoted as VTW) (Lin et al., 2025) under the same pruning ratio. According to Table 1, removing all visual tokens at a fixed layer results in suboptimal performance on the TextVQA benchmark (DART + Random pruning achieves 82.7% accuracy compared to 76.3% accuracy for DART + VTW). This trend is evident across other visually complex benchmarks, which depend on information from deeper layers. For example, DivPrune + Random pruning achieves 2.6% and 3.0% higher accuracy compared to the VTW counterpart on DocVQA and OCRBench, respectively.

## 5 CONCLUSION

In this paper, we investigate the visual token information in VLLMs, which is defined as the change of output probability when a visual token is removed. Our experimental findings reveal that visual token information diminishes to almost zero at a particular layer, influenced by task visual complexity and model visual capability. This explains why current token pruning methods do not surpass random pruning at deep layers, showing that integrating random pruning yields superior results across various models and benchmarks. We believe our findings offer empirical insights for employing informative visual tokens to enhance VLLMs.

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

## A EXPERIMENT DETAILS

**Datasets.** We conduct experiments on nine widely adopted benchmarks, which we categorize into three groups to reflect varying levels of visual complexity: *(i) knowledge QA*, including GQA (Hudson & Manning, 2019), SQA (ScienceQA) (Lu et al., 2022), and POPE (Li et al., 2023), which require factual reasoning grounded in visual and textual information; *(ii) text-centric understanding* , including VQA$^{\text{Text}}$ (TextVQA) (Singh et al., 2019), InfoVQA (Mathew et al., 2022), DocVQA (Mathew et al., 2021), and OCRBench (Liu et al., 2024d), which involve extracting and reasoning over fine-grained text in images, leading to higher visual complexity; *(iii) Comprehensive evaluation*, including MME (Fu et al., 2024), MMB (MMBench) and MMB-CN (MMBench-CN) (Liu et al., 2024c), which cover a broad range of multimodal tasks and provide a comprehensive assessment of a model's multimodal understanding capability.

**VLLM Architectures.** We verify different pruning strategies on two representative VLLM architectures: LLaVA-1.5 (Liu et al., 2024a), which decodes an image into a fixed number of visual tokens, and Qwen2.5-VL (Bai et al., 2025), which adaptively determines the number of visual tokens based on the resolution of the input image.

**Combining existing methods with random pruning.** DivPrune removes visual tokens before they enter the language decoder, while DART and FastV perform pruning at the second and third layers, respectively. We study two hybrid pruning strategies: first, we apply $S \in \{\text{FastV}, \text{DART}, \text{DivPrune}\}$ at a shallow layer to prune part of the visual tokens. Subsequently, at a specified deep layer, an additional pruning step is performed: (i) $S$+VTW: following VTW (Lin et al., 2025), all remaining visual tokens are withdrawn, or (ii) $S$+Random: random pruning is applied to further remove visual tokens. Detailed pruning layers and ratios are illustrated in Table 3 and Table 4. For pruning on Qwen2.5-VL-7B (28 decoder layers, with an average per-layer retention of 50% visual tokens) and LLaVA-1.5-7B (32 decoder layers, with an average per-layer retention of 192 visual tokens), we follow the original methods for pruning at their respective shallow layers, while applying an additional pruning step at deeper layers: $S$+VTW prunes all tokens at the 26th and 21th layers for Qwen and LLaVA, respectively, whereas $S$+Random applies random pruning at the 25th and 20th layers. For pruning on LLaVA-1.5-7B (with an average per-layer retention of 64 visual tokens), pruning at the second or third layer would retain too few tokens under this setting. For instance, DART need to prune 89% of visual tokens while FastV need to prune 98%. Therefore, for the hybrid strategies based on DART and FastV, the shallow pruning layer is shifted one layer earlier to prevent pruning away an excessive number of tokens at the shallow pruning stage, while the shallow pruning layer remains is unchanged for hybrid strategies based on DivPrune.

Table 3: Pruning settings for Qwen2.5-VL-7B (28 decoder layers) with an average per-layer retention of 50% visual tokens. Layer 0 indicates pruning before the visual tokens enter the language decoder.

| Retain 50% Tokens (↓ 50%) | | |
|---|---|---|
| Method | Layer | Retain ratio |
| DART | 2 | 0.46 |
| DART+VTW | 2 | 0.49 |
| | 26 | 0 |
| DART+Random | 2 | 0.49 |
| | 25 | 0.25 |
| DivPrune | 0 | 0.50 |
| DivPrune+VTW | 0 | 0.53 |
| | 26 | 0 |
| DivPrune+Random | 0 | 0.53 |
| | 25 | 0.25 |

## B LLM USAGE STATEMENT

We confirm that no large language models (LLMs) were used at any stage of this work, including research ideation, experimental design, data analysis, or manuscript writing. All ideas, methods,

Table 4: Pruning settings for LLaVA-1.5-7B (32 decoder layers) with an average per-layer retention of 192 and 64 visual tokens.

| Method | Retain 192 Tokens (↓ 66.7%) | | Retain 64 Tokens (↓ 88.9%) | |
| | Layer | Retain ratio | Layer | Retain ratio |
|---|---|---|---|---|
| DART | 2 | 0.29 | 2 | 0.05 |
| DART+VTW | 2 | 0.44 | 1 | 0.10 |
| | 21 | 0 | 26 | 0 |
| DART+Random | 2 | 0.44 | 1 | 0.10 |
| | 20 | 0.07 | 20 | 0.05 |
| DivPrune | 0 | 0.33 | 0 | 0.11 |
| DivPrune +VTW | 0 | 0.49 | 0 | 0.17 |
| | 21 | 0 | 21 | 0 |
| DivPrune +Random | 0 | 0.49 | 0 | 0.17 |
| | 20 | 0.07 | 20 | 0.02 |
| FastV | 3 | 0.26 | 3 | 0.02 |
| FastV+VTW | 3 | 0.41 | 2 | 0.06 |
| | 21 | 0 | 26 | 0 |
| FastV+Random | 3 | 0.41 | 2 | 0.06 |
| | 20 | 0.06 | 20 | 0.03 |

and results were conceived and executed entirely by the authors, who take full responsibility for the content of this paper.

