# OpenReview forum: "All You Need Are Random Visual Tokens?  Demystifying Token Pruning in VLLMs"
_ICLR.cc/2026/Conference — ICLR 2026 Conference Withdrawn Submission_

### Official Review · Reviewer_yk93 · 2025-10-30

**Soundness:** 3
**Presentation:** 3
**Contribution:** 3
**Rating:** 6
**Confidence:** 4

**Summary:**

This paper investigates token pruning in Vision-Language Large Models (VLLMs) and introduces the concept of an "information horizon"—a layer beyond which visual tokens contribute little to model output. The authors propose an information score based on output perturbation to quantify token importance, and use it to analyze the salience of visual tokens across layers. They find that simple random pruning in deep layers can balance efficiency and performance, and further enhance existing pruning strategies.

**Strengths:**

1. The paper presents a compelling empirical finding—the existence of an “information horizon” in deeper layers of VLLMs—where visual tokens become increasingly redundant.

2. The proposed method for quantifying token information via output perturbation is intuitive and well-validated.

**Weaknesses:**

1. The paper proposes to use information scores to identify the “information horizon,” beyond which pruning can be safely applied. However, calculating this score for every visual token requires measuring perturbations in output probability, which can be computationally expensive.

2. The finding that pruning visual tokens in deeper layers does not significantly degrade performance raises the question of whether essential information has already been transferred to the text tokens. However, the paper does not investigate the information content of text tokens. Analyzing their information scores might uncover further insights into modality interactions or model reliance on textual representations in later stages.

3. Since pruning is applied only in the final few layers, the efficiency improvements may be limited. The paper would benefit from a more detailed analysis of actual computational savings, such as FLOPs reduction or latency measurements. Furthermore, the overhead of computing the optimal pruning layer could negate some of these efficiency gains, making the approach more observational than practically deployable.

4. While the paper empirically observes the existence of an information horizon, it lacks an in-depth explanation of why visual token information diminishes in deeper layers. A more detailed exploration could enhance the scientific contribution of the work.

**Questions:**

1. While the paper demonstrates that random pruning in deeper layers is effective, it remains unclear how the specific layers for pruning are selected in the experimental setup or tables. Is the pruning depth fixed across all downstream tasks, or is it adjusted for each task individually?

---

### Official Review · Reviewer_YVyB · 2025-10-31

**Soundness:** 3
**Presentation:** 3
**Contribution:** 2
**Rating:** 4
**Confidence:** 4

**Summary:**

This paper studies token pruning in Vision-Language Large Models (VLLMs) and discovers that, in deeper layers, existing pruning methods perform no better than random pruning. The authors propose measuring each visual token’s “information” by the change in output probability upon its removal, revealing that token information gradually vanishes at an “information horizon.” This horizon varies with task complexity and model capability. It demonstrates that simply integrating random pruning with existing pruning methods can more effectively balance inference efficiency and accuracy across various datasets

**Strengths:**

1. The paper is overall clearly written and easy to follow.
2. It thoroughly investigates the behavior of visual information in the deeper layers of multimodal large models.

**Weaknesses:**

1. The results shown in the teaser are not very convincing. The three benchmarks used are either too simple or yield unstable evaluations. It would be more appropriate to choose datasets such as TextVQA, MMBench, or DocVQA instead.

2. The overall observation seems quite similar to that in PDrop. A discussion and comparison with the observations in PDrop should be added to the paper.

3. When estimating a visual token’s information in shallow layers, multiple forward passes are required, which incurs very high computational cost.

**Questions:**

see weaknesses

---

### Official Review · Reviewer_nD3h · 2025-10-31

**Soundness:** 2
**Presentation:** 2
**Contribution:** 1
**Rating:** 2
**Confidence:** 4

**Summary:**

This paper investigates the phenomenon of vanishing token information, observing that visual tokens become increasingly redundant in intermediate and deeper layers. Building on this insight, the authors propose a simple random pruning strategy applied to deeper layers to balance the trade-off between performance and efficiency. Extensive experiments across various models and benchmarks demonstrate that integrating existing methods with random pruning yields state-of-the-art results.

**Strengths:**

1. This paper presents an analysis showing that visual token information becomes increasingly trivial in deeper layers.
2. Experimental results demonstrate that integrating existing approaches with random token pruning leads to performance improvements.

**Weaknesses:**

1. The novelty of this paper is limited. Prior studies have already observed that visual tokens become less important in deeper layers [1], and another work [2] has examined the effects of random token pruning. VScan [3] has also shown that model predictions tend to converge at varying intermediate layers; therefore, performing token pruning in deeper layers can effectively balance efficiency and performance.
2. The performance gains are relatively minor, with improvements of less than 0.5%.
3. The experimental results are not comprehensive. For example, the paper does not report performance on video question answering tasks. In addition, comparisons with more recent baselines are missing.
4. This work reads more like a technical report rather than a research paper. It primarily presents empirical observations showing that applying random pruning can yield slight improvements, but lacks deeper theoretical analysis or novel methodological insights to substantiate or generalize the findings.
5. It is unclear how the efficiency of the proposed approach compares with existing methods. A detailed comparison in terms of computational cost, inference speed would strengthen the paper.

[1] PyramidDrop: Accelerating Your Large Vision-Language Models via Pyramid Visual Redundancy Reduction. https://arxiv.org/abs/2410.17247.

[2] Token Pruning in Multimodal Large Language Models: Are We Solving the Right Problem? https://www.arxiv.org/abs/2502.11501.

[3] VScan: Rethinking Visual Token Reduction for Efficient Large Vision-Language Models. https://arxiv.org/abs/2505.22654.

**Questions:**

See the weaknesses section above.

---

### Official Review · Reviewer_6ECg · 2025-11-01

**Soundness:** 2
**Presentation:** 3
**Contribution:** 2
**Rating:** 4
**Confidence:** 4

**Summary:**

This paper presented a study of visual token pruning mechanism in VLMs. The authors first proposed a metric called visual token information (layer specific), which is defined as the contribution of a specific visual token to the ground truth output logits, and it is measured by passing only that visual token after a certain layer and compare against no visual token baseline. From this new metric definition, the analysis of several visual token pruning methods shows that their selection of tokens exhibit no difference than random selection at deeper layers, i.e. in deeper layers, all visual tokens contain very similar information. Based on this finding, they proposed to simply drop a random subset of visual tokens in deep layers to improve efficiency while keeping performance. The proposed random dropping can also be combined with existing token pruning methods to achieve better performance.

**Strengths:**

This paper presented some interesting analysis of how different visual token pruning methods’s token selection differs in early layers but exhibit much larger similarity in deeper layers.

The proposed random token dropping in deeper layers, when combined with existing visual token pruning methods, yield further improvements on 7 visual understanding benchmarks using two VLMs.

**Weaknesses:**

The overall novelty and contribution of this paper feels quite limited. The proposed visual token information measurement is actually a commonly used technique to measure the important of individual tokens (Jain and Wallace 2019), the insights about how shallow layers contains much more interaction between visual and text tokens but deeper layers require much less interaction have been discussed in previous works such as VTW (Lin et al. 2024) and VScan (Zhang et al. 2025). In fact, the proposed random token dropping is very similar to VTW conceptually except that a small random subset of the tokens are still kept in this paper.

Sarthak Jain and Byron C. Wallace. 2019. Attention is not Explanation. In Proceedings of the 2019 Conference of the North American Chapter of the Association for Computational Linguistics: Human Language Technologies, Volume 1 (Long and Short Papers), pages 3543–3556, Minneapolis, Minnesota. Association for Computational Linguistics.

Lin, Zhihang, Mingbao Lin, Luxi Lin and Rongrong Ji. “Boosting Multimodal Large Language Models with Visual Tokens Withdrawal for Rapid Inference.” ArXiv abs/2405.05803 (2024): n. pag.

Zhang, Ce, Kaixin Ma, Tianqing Fang, Wenhao Yu, Hongming Zhang, Zhisong Zhang, Yaqi Xie, Katia P. Sycara, Haitao Mi and Dong Yu. “VScan: Rethinking Visual Token Reduction for Efficient Large Vision-Language Models.” ArXiv abs/2505.22654 (2025): n. pag.

**Questions:**

Have you tried the same analysis on larger models? e.g. 72B scale where the number of layers are much larger. Will the same conclusion still hold?

---

### Note · Authors · 2025-11-14

I have read and agree with the venue's withdrawal policy on behalf of myself and my co-authors.